# A contact binary satellite of the asteroid (152830) Dinkinesh

Asteroids with diameters less than about 5 km have complex histories because they are small enough for radiative torques (that is, YORP, short for the Yarkovsky–O'Keefe–Radzievskii–Paddack effect)[1] to be a notable factor in their evolution[2]. (152830) Dinkinesh is a small asteroid orbiting the Sun near the inner edge of the main asteroid belt with a heliocentric semimajor axis of 2.19 AU; its S-type spectrum[3,4] is typical of bodies in this part of the main belt[5]. Here we report observations by the Lucy spacecraft[6,7] as it passed within 431 km of Dinkinesh. Lucy revealed Dinkinesh, which has an effective diameter of only 720 m, to be unexpectedly complex. Of particular note is the presence of a prominent longitudinal trough overlain by a substantial equatorial ridge and the discovery of the first confirmed contact binary satellite, now named (152830) Dinkinesh I Selam. Selam consists of two near-equal-sized lobes with diameters of 210 m and 230 m. It orbits Dinkinesh at a distance of 3.1 km with an orbital period of about 52.7 h and is tidally locked. The dynamical state, angular momentum and geomorphologic observations of the system lead us to infer that the ridge and trough of Dinkinesh are probably the result of mass failure resulting from spin-up by YORP followed by the partial reaccretion of the shed material. Selam probably accreted from material shed by this event.

Dinkinesh was a late addition to the Lucy mission and was intended primarily as an in-flight test of an autonomous range-finding and tracking system that is a critical component of Lucy's operations[7]. It was an appealing target because the fly-by geometry closely mimicked that of the Trojan targets to be encountered later in the mission[6]. Lucy approached Dinkinesh at a solar phase angle of 120°; at close approach, the phase decreased rapidly, passed through near-zero and then increased to an outbound phase of 60°. The relative velocity of Lucy and Dinkinesh was 4.5 km s⁻¹. At closest approach, Lucy was 430.629 ± 0.045 km from Dinkinesh and had a Lucy–Dinkinesh–Sun angle of 30°. A sample of the high-resolution images is shown in Fig. 1. The basic shape of Dinkinesh is reminiscent of the 'top' shapes seen in the near-Earth asteroid (NEA) population (for example, Moshup[8], Bennu[9] Ryugu[10] and—to a lesser extent—Didymos[11,12]). Dinkinesh is similarly sized as well. As described in more detail below, Dinkinesh has an effective diameter of 719 m, whereas Bennu, Ryugu and Didymos have effective diameters of between approximately 560 m and 900 m. Like these objects, Dinkinesh is dominated by a prominent equatorial ridge. Dinkinesh also has a large trough running nearly perpendicular to the ridge. Although both Ryugu and Didymos have similar features[13] (O. S. Barnouin et al., manuscript in preparation), the trough on Dinkinesh seems to be more substantial. The ridge overlays the trough, implying that it is the younger of the two structures. However, there is no information on their absolute ages and thus they could potentially have formed in the same event.

High-resolution images obtained throughout the encounter (see the 'Observations' section in Methods) make it possible to reconstruct shape models for each of the components. Owing to the small size of Dinkinesh and Selam, usefully resolved imaging was possible for only several minutes before and after close encounter. The rotation of

Dinkinesh was observed, but the amount of extra terrain revealed by the rotation was small (approximately 10%) compared with the unilluminated portion of the body. No rotational or orbital motion of Selam was seen. Illumination of the anti-solar hemisphere of Dinkinesh from Selam was too faint to be observed. Thus only one hemisphere of each body is visible in imaging. However, constraints on the unobserved hemispheres can be provided by photometry from both the ground[14] and Lucy when it was too far away to resolve the targets. We therefore turn our attention to the analysis of this photometry before we further discuss the shapes and structure of the system.

The unresolved data from the post-encounter light-curve photometry campaign (see the 'Observations' section in Methods) is described in Fig. 2. From these data, we determine that the contribution of Selam's rotation to the light curve has periodicity with $T = 52.44 \pm 0.14$ h, comparable with the $52.67 \pm 0.04$ h period found from ground-based observations[14]. We adopt the ground-based period of 52.67 h because it is more precise owing to its longer sampling baseline. The post-encounter light curve also shows dips inferred to be because of mutual eclipses of Dinkinesh and Selam with the same 52-h periodicity (Fig. 2 and the 'Observations' section in Methods), demonstrating that the orbital period of Selam is very similar to its rotational period. We interpret this to mean that the system is tidally locked. By using the formalism in ref. 15, we estimate that the timescale for tidal effects to align the long axis of Selam radially relative to Dinkinesh to be short, on the order of $10^5$ years at the current separation, although their formalism might not be accurate because some important radiation effects[1] were not considered[16]. We also find that the centres of Dinkinesh and the two lobes of Selam seem to lie along a single line (Fig. 1m)—consistent with a tidally locked system. Thus we conclude that Selam is in synchronous rotation and thereby orbits Dinkinesh with a period of 52.67 h.

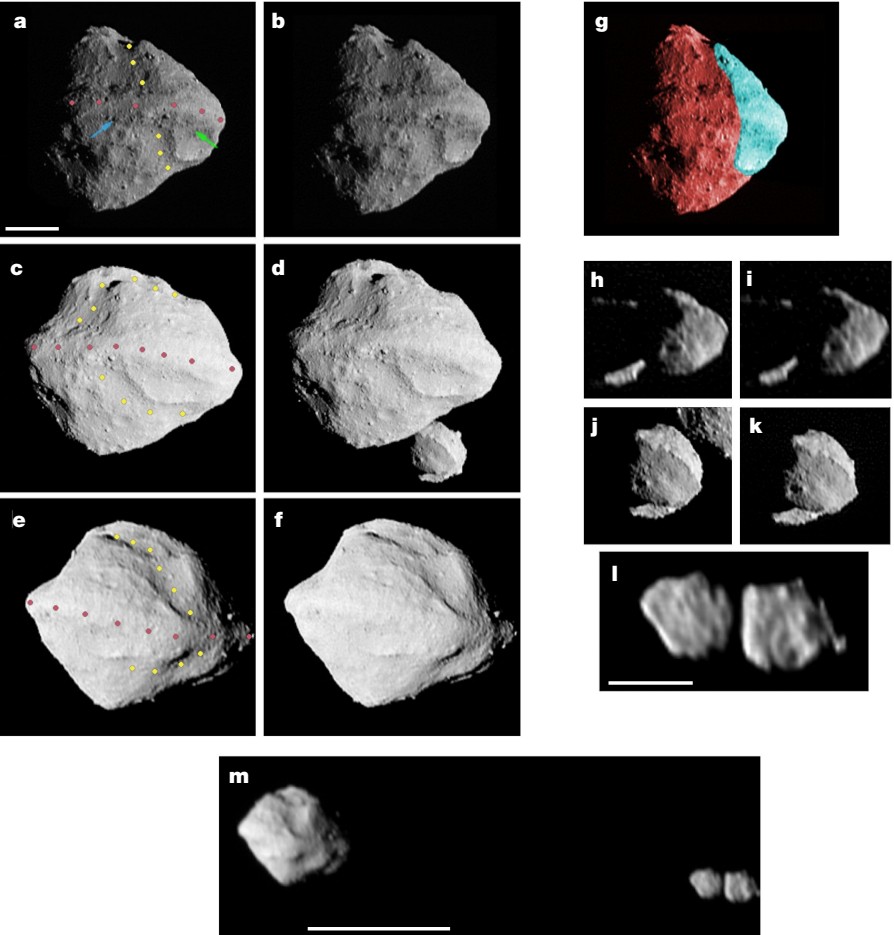

**Fig. 1 | Images of Dinkinesh and Selam obtained by Lucy's close-approach imaging campaign. a–f,** Cross-eyed stereo versions of the images taken on approach, near-close approach and on departure, respectively (see the 'Observations' section in Methods for a description of the imaging campaign). Dinkinesh has two main geological features: a longitudinal trough and an equatorial ridge (the yellow and rose coloured dots, respectively). The two coloured arrows (green and blue) in **a** point to the northern boundary of the ridge, as determined by visual inspection, at the two fiducial longitudes discussed in Fig. 3. Scale bar, 200 m. **g,** A simulated image of Dinkinesh with the trough removed. This is a modified version of **a**, in which the cyan region was moved 79 m to the lower left in the image (26° from horizontal) and rotated 7° clockwise. We take the fact that the limb profile of Dinkinesh is smooth near the colour transitions of this reconstruction to suggest that the trough is a result of a structural failure that moved the cyan region away from the remainder of the body. **h–k,** Stereo pairs of images of Selam taken on approach and near-close approach, respectively. **l,** A single image of Selam taken on departure.

Selam was outside the L'LORRI field of view from 10 s to 5.5 min after close approach and so stereo imaging is not possible. The images of Selam allow us to visually estimate the dimensions of its lobes by crudely approximating their complex shapes as triaxial ellipsoids. We find that the inner and outer lobe major axes lengths in the directions parallel to the Dinkinesh vector, the orbital direction and the spin pole are roughly 240 × 200 × 200 m and 280 × 220 × 210 m, respectively. Scale bar, 200 m. **m,** A departure image of the entire system. Scale bar, 1 km. Also, all images are deconvolved except for **m. l** and **m** are the same image, so comparison illustrates the effects of deconvolution. Ecliptic north is approximately up in all frames, whereas the body north of Dinkinesh is down because it is a retrograde rotator. Image details are as follows. Times relative to close approach in minutes: **a**, −1.04; **b**, −1.29; **c**, +0.21; **d**, −0.04; **e**, +2.21; **f**, +1.71; **h**, −2.29; **i**, −3.29; **j**, −0.29; **k**, −0.54; **l** and **m**, +5.46. Original pixel scale, m per pixel: **a**, 2.53; **b**, 2.72; **c**, 2.14; **d**, 2.12; **e**, 3.63; **f**, 3.11; **h**, 3.70; **i**, 4.85; **j**, 2.16; **k**, 2.24; **l** and **m**, 7.56. Solar phase angle, °: **a**, 62.1; **b**, 68.0; **c**, 21.5; **d**, 30.5; **e**, 25.0; **f**, 17.8; **h**, 84.2; **i**, 93.3; **j**, 39.3; **k**, 47.7; **l** and **m**, 44.5.

The timing of the mutual events in the post-encounter light curve (Fig. 2), relative to the orbital position of Selam during the fly-by, shows that the orbit of Selam must be retrograde with respect to the heliocentric orbit of Dinkinesh.

The primary, Dinkinesh, rotates more rapidly, with the best fit to the light curve giving a spin period of $P = 3.7387 \pm 0.0013$ h. Feature tracking during the fly-by shows that the rotation is retrograde with respect to ecliptic north, that is, in the same sense as the orbit of Selam. The overall spin state (a synchronous secondary and a rapidly spinning primary) makes Dinkinesh similar to most small near-Earth and main-belt asteroids with close satellites[17].

We now return to the topic of the shapes of Dinkinesh and Selam. A model of Dinkinesh produced by the process described in the 'Shape'

section in Methods and based on a preliminary reconstruction of the trajectory of Lucy is illustrated in Fig. 3. We find a volume-equivalent spherical diameter of $719 \pm 24$ m for Dinkinesh based on this shape model. Selam seems to consist of two distinct lobes. However, the contact point was in shadow during the encounter and so the exact nature of the neck is uncertain. Images taken during approach in which the outer lobe was farther away from the spacecraft than the inner lobe (see Fig. 1h for example) show that the neck is less than about 67% of the diameter of the inner lobe. We find equivalent spherical diameters of $212 \pm 21$ m and $234 \pm 23$ m for the inner and outer lobes of Selam based on fitting ellipses to visual limb profiles. If the lobes were in orbit about one another, their period would be about 4 h, which is inconsistent with the light-curve observations described above.

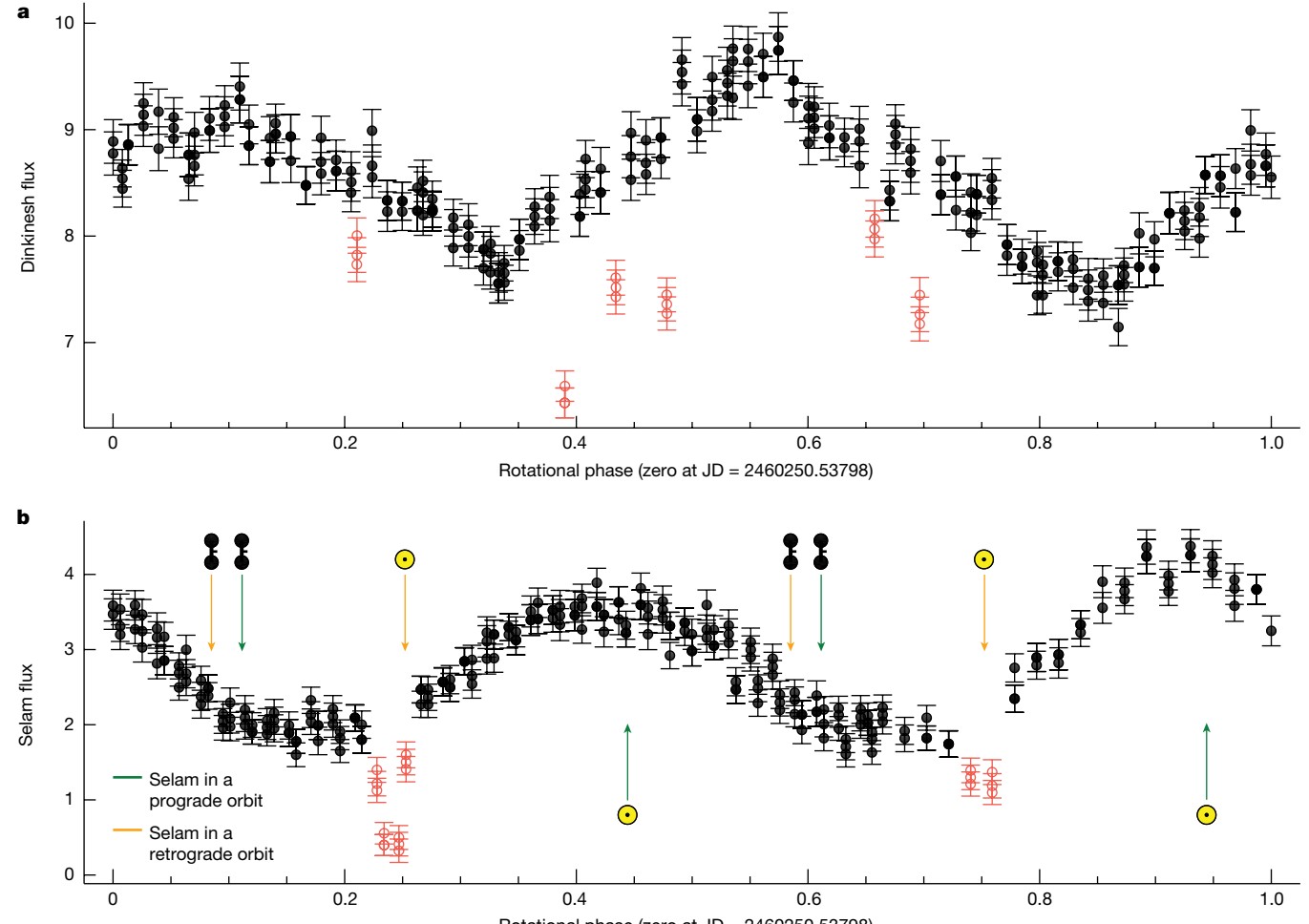

**Fig. 2 | Phased light curves for Dinkinesh and Selam. a**, Phased light curve for Dinkinesh folded using a period of 3.7387 h. **b**, Phased light curve for Selam folded using a period of 52.67 h. These periods were determined from outbound photometry, as developed in the 'Light-curve analysis' section in Methods. The raw photometry is shown in Extended Data Fig. 1. The solid black points were used to derive the periods. The light curve of Dinkinesh is more complicated than that of Selam. Indeed, the light curve of Selam is reminiscent of what is expected for a contact binary consisting of two rotating spheroids seen edge-on and at this phase angle (60°)[30]. The hollow red points were excluded and correspond to mutual events. The arrows indicate when different types of event would be predicted. Events marked with the Lucy spacecraft symbol, ●●, show occultations (when one object passes in front of the other from the point of view of the spacecraft) if they occur. Events marked with the sun symbol, ⊙, indicate the potential times of eclipses (at which the shadow of one object falls on the other). The observed mutual events are associated with eclipses. Occultations are not seen by Lucy during departure, which is consistent with the fact that its trajectory is slightly inclined with respect to the orbital plane of Dinkinesh. Green arrows show events that occur if Selam were in a prograde orbit about Dinkinesh, whereas orange arrows occur for a retrograde orbit. From this, we can conclude that the orbit of Selam is retrograde.

Furthermore, we would have detected motion if the period was that short. Thus the lobes must be resting on one another and Selam is probably a contact binary.

Outbound images clearly show both lobes of Selam (Fig. 1m) from a direction almost perpendicular to the vector between them, as determined by triangulation. From these images, we derive a preliminary estimate of the centre-of-figure separation between Dinkinesh and Selam to be 3.11 ± 0.05 km at the time of the fly-by. We argue in the 'Mass and density' section in Methods that Selam is on a circular orbit. If so, this separation represents the semimajor axis of the mutual orbit.

The orientation in space of both Dinkinesh and Selam can be estimated with current data. In particular for Dinkinesh, the small amount of rotation observed during the encounter and the direction of its shape model's short axis suggests that its obliquity is approximately 178.7 ± 0.5° (that is, its rotational axis is about 1° from being perpendicular to its orbital plane). For the satellite, the mutual eclipses observed during the post-encounter Lucy observations, and mutual events inferred from the 2022–2023 ground-based light curve[6], suggest that its orbit plane is close to the heliocentric orbital plane of Dinkinesh. It is therefore likely that all three, the heliocentric orbit of Dinkinesh, the orbit of Selam and the equatorial plane of Dinkinesh, are close to one another. This configuration is nearly ubiquitous among small binary asteroids[18] as a result of spin-pole reorientation by the asymmetric thermal radiation forces caused by the YORP effect[1]. The YORP timescale is less than about 10[7] years for the spin-pole of Dinkinesh to approach either zero or 180° (refs. 19,20).

The inner lobe of Selam also has a prominent ridge-like structure (Fig. 1h–k). Both lobes of Selam have flat facets and a blocky, angular overall shape, and the apparent ridge may be the boundary of two such facets. If, however, the structure formed from the accretion of material from a Dinkinesh-centred disk, as one might expect, it would have originally been aligned with both the orbit plane and the ridge of Dinkinesh. In that case, it is likely that the ridge of Selam then became misaligned

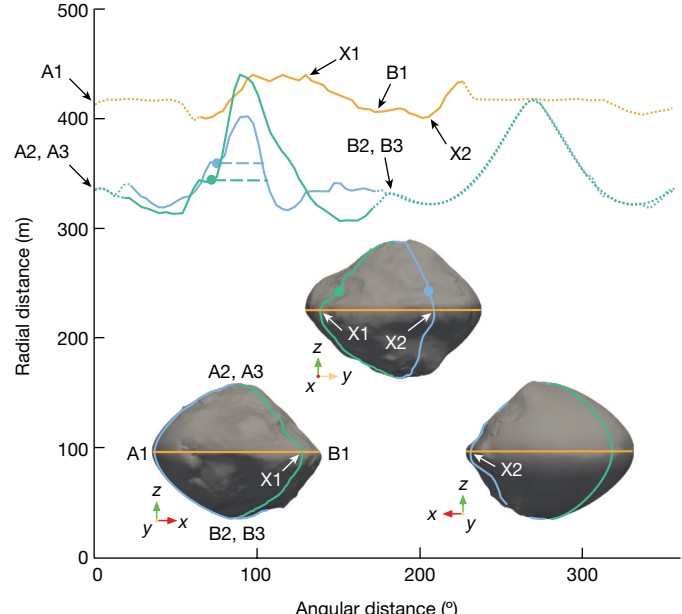

**Fig. 3 | The shape model of Dinkinesh.** Three orientations (insets) and topographic cuts designed to emphasize the structure of the equatorial ridge. The model, which is described in the 'Shape' section in Methods, consists of two regions. The side of the model that was facing Lucy during the encounter was based on the images taken during the close-approach imaging campaign (see the 'Observations' section in Methods). The unilluminated portion of Dinkinesh is estimated with a super-ellipsoid. The rotational ($z$) axis points up in these figures. The orange curve shows a latitudinal cut that lies along the ridge. The blue and green curves are longitudinal cuts corresponding to the minimum and maximum elevations of the equatorial ridge, respectively. The points labelled X1 and X2 indicate where the green and blue curves cross the orange curve, respectively. The dots show the location of the ridge's northern boundary, as determined visually in Fig. 1a (the corresponding blue and green arrows) and the horizontal dashed lines show the extent of the ridge. The ridge at its lowest point measures 150 m wide and 40 m high, whereas it is 230 m wide and 100 m high at its highest point. The curves are solid in the locations in which the shape model is reliable, whereas they are dotted elsewhere. The reference locations labelled A1, A2, A3, B1, B2, B3, X1 and X2 are included to allow the reader to associate the shape model to the curves.

during the formation of the contact binary, but this implies that either (1) Selam is at present rotating or librating about its long axis or (2) its ridge formed before contact. The observed structure of Selam implies that it is a rubble pile, at least partially. However, the angular, binary shape of Selam implies substantial internal strength and is substantially different from the oblate spheroid shape of Dimorphos, the moon of Didymos[21], the only other satellite of a sub-kilometre asteroid (also an S-type) for which we have detailed images.

The mineralogy and bulk density of Dinkinesh provide constraints on its structure. The bulk density of Dinkinesh is 2,400 ± 350 kg m$^{-3}$ ('Mass and density' section in Methods), which is in the range of expected values for objects with ordinary chondrite mineralogies. Bulk densities of L-chondrite meteorites, which are a good analogue for the range of ordinary chondrites[22] and have the expected mineralogy for S-type asteroids, average 3,360 ± 160 kg m$^{-3}$ with 7.5% microporosity[23]. Given the uncertainties in Dinkinesh bulk density discussed in the 'Mass and density' section in Methods, this suggests a macroporosity of 25 ± 10%. Its bulk density is consistent with the S-type NEAs of this mineralogy and in this size range. For example, although Didymos has a similar density of 2,800 ± 280 kg m$^{-3}$ (O. S. Barnouin et al., manuscript in preparation), the bulk density of Itokawa is 1,900 ± 130 kg m$^{-3}$ (ref. 24) and that of the radar-observed binary Moshup is 1,970 ± 240 kg m$^{-3}$ (ref. 8).

The low-density objects are probably much more porous and have a more pronounced rubble-pile structure than Didymos, with Dinkinesh some place in between. Dinkinesh and Didymos are probably on part of a continuum in which substantial portions of the object are relatively coherent.

Dinkinesh accounts for 94% of the volume of the system, with Selam accounting for 6%. If we assume that all of the components have an equal density, the component masses of Dinkinesh and Selam, are $M_D = 4.67 \times 10^{11}$ and $M_S = 0.28 \times 10^{11}$ kg, respectively. Using these component masses, it is possible to calculate that the barycentre is offset from the centre of mass of Dinkinesh by a distance $s_{bary} = 176$ m in the direction of Selam, well interior to the body of the primary.

Figure 1 strongly suggests that Dinkinesh suffered a global structural failure in its past. Given its small size, this event is probably the result of spin-up by the YORP effect[1,25]; see discussion in Fig. 4 caption. If true, then the angular momentum of the Dinkinesh system should be comparable with the total angular momentum of a parent body spinning near the spin-barrier limit[26]. Indeed, we find that the Dinkinesh system contains 88% of the angular momentum required for rotational break-up (see the 'Angular momentum' section in Methods), which is consistent with the idea that the structure of Dinkinesh failed owing to its large angular momentum.

Dinkinesh shares many characteristics with other similar-sized asteroids, both near-Earth and main belt, and is the only sub-kilometre-sized main-belt object ever studied at close range. Approximately 15% of small asteroids are observed to be binaries[18,27]. For the subset of these systems that are well characterized, the dominant pattern is a system with a synchronous secondary in a near-circular orbit with a semimajor axis, $a$, of approximately 3 or more primary radii, $r_{prim}$ (ref. 27). The semimajor axis of Selam, at $a/r_{prim} \approx 9$, is wide compared with most other well-characterized systems of similar size that cluster closer to $a/r_{prim} \approx 3$ (ref. 28). The spin period of Dinkinesh is also longer than the approximately 2.5-h period typically observed in the NEA binary population[27]. One possible scenario is that Selam originally formed nearer to Dinkinesh and then evolved to a larger semimajor axis through tidal interaction and/or binary YORP that also slowed down the rotation of Dinkinesh[29].

The most distinctive characteristic of the Dinkinesh–Selam system is the contact binary structure of Selam. Figure 4 shows three possible scenarios for its formation. The binary nature of Selam places important constraints on the formation of these satellite systems no matter how it formed. First, the fact that the two lobes are nearly the same diameter argues that the satellite-formation process responsible for Selam favours building objects of a particular size. As far as we are aware, none of the formation models in the literature has been shown to meet this requirement. Second, as we describe above, the two lobes are distinct bodies, so the process that brought the two lobes together must have done so with a small enough velocity for the lobes to have survived.

The unexpected complexity of the Dinkinesh system strongly suggests that small asteroids in the main belt are more complex than previously thought. The fact that a contact binary can form in orbit about a larger object suggests a new mode for the formation of small bilobed bodies such as Itokawa[24], for which they may once have been components of a system such as Dinkinesh that subsequently became unbound.

## Online content

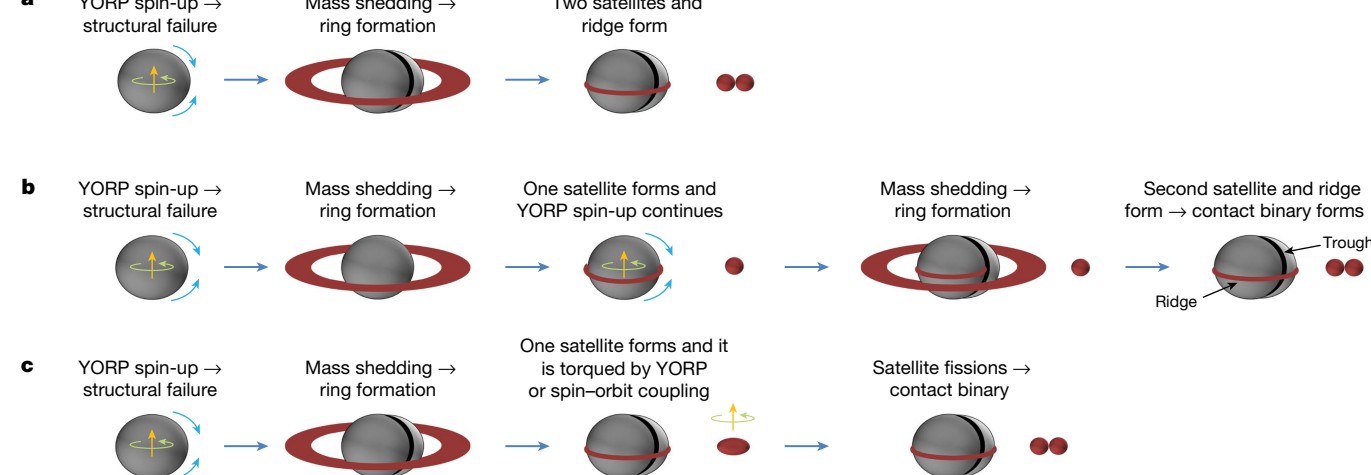

**a** YORP spin-up → structural failure   Mass shedding → ring formation   Two satellites and ridge form

**b** YORP spin-up → structural failure   Mass shedding → ring formation   One satellite forms and YORP spin-up continues   Mass shedding → ring formation   Second satellite and ridge form → contact binary forms — Trough — Ridge

**c** YORP spin-up → structural failure   Mass shedding → ring formation   One satellite forms and it is torqued by YORP or spin–orbit coupling   Satellite fissions → contact binary

**Fig. 4 | A simplified graphical depiction of a plausible sequence of events leading to the current configuration of the Dinkinesh–Selam system. a,b**, Asteroids with diameters less than approximately 10 km are subject to spin-up by the YORP effect[1]. Rapid spin of the primary and the associated centrifugal force eventually trigger a structural failure that leads to sudden mass shedding[25]. This event might also have created the trough seen on Dinkinesh (Fig. 1a–f) through the mass movement of a portion of the body (Fig. 1g). The shed material forms a ring, with some material coalescing into a satellite(s) and closer material eventually falling back to the surface at the equator to form the ridge[31]. The formation of the contact binary may be the result of a merger of two satellites formed either in a single mass-shedding event (**a**) or in two separate events (**b**)[32]. **c**, An alternative scenario is that Selam formed as a single object that subsequently underwent fission owing to spin–orbit coupling[15,33]. It is also possible that some or all of Selam formed from a collision on the primary[34], but the trough and ridge would not have survived such an event. Thus this mechanism would still require later Dinkinesh spin-up by YORP and mass shedding to form the trough and superposed equatorial ridge.

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

**Harold F. Levison**[1✉]**, Simone Marchi**[1]**, Keith S. Noll**[2]**, John R. Spencer**[1]**, Thomas S. Statler**[3]**, James F. Bell III**[4]**, Edward B. Bierhaus**[5]**, Richard Binzel**[6]**, William F. Bottke**[1]**, Daniel Britt**[7]**, Michael E. Brown**[8]**, Marc W. Buie**[1]**, Philip R. Christensen**[4]**, Neil Dello Russo**[9]**, Joshua P. Emery**[10]**, William M. Grundy**[10,11]**, Matthias Hahn**[12]**, Victoria E. Hamilton**[1]**,**

Carly Howett[13], Hannah Kaplan[2], Katherine Kretke[1], Tod R. Lauer[14], Claudia Manzoni[15], Raphael Marschall[16], Audrey C. Martin[7], Brian H. May[15], Stefano Mottola[17], Catherine B. Olkin[18], Martin Pätzold[12], Joel Wm. Parker[1], Simon Porter[1], Frank Preusker[17], Silvia Protopapa[1], Dennis C. Reuter[2], Stuart J. Robbins[1], Julien Salmon[1], Amy A. Simon[2], S. Alan Stern[1], Jessica M. Sunshine[19], Ian Wong[2,20], Harold A. Weaver[9], Coralie Adam[21], Shanti Ancheta[5], John Andrews[1], Saadat Anwar[4], Olivier S. Barnouin[9], Matthew Beasley[1], Kevin E. Berry[2], Emma Birath[1], Bryce Bolin[2], Mark Booco[5], Rich Burns[2], Pam Campbell[5], Russell Carpenter[2], Katherine Crombie[22], Mark Effertz[5], Emily Eifert[5], Caroline Ellis[5], Preston Faiks[5], Joel Fischetti[21], Paul Fleming[23], Kristen Francis[5], Ray Franco[5], Sandy Freund[5], Claire Gallagher[5], Jeroen Geeraert[21], Caden Gobat[1], Donovan Gorgas[5], Chris Granat[5], Sheila Gray[5], Patrick Haas[5], Ann Harch[24], Katie Hegedus[5], Chris Isabelle[5], Bill Jackson[5], Taylor Jacob[5], Sherry Jennings[25], David Kaufmann[1], Brian A. Keeney[1], Thomas Kennedy[5], Karl Lauffer[26], Erik Lessac-Chenen[21], Rob Leonard[27], Andrew Levine[21], Allen Lunsford[20], Tim Martin[5], Jim McAdams[21], Greg Mehall[4], Trevor Merkley[5], Graham Miller[5], Matthew Montanaro[28], Anna Montgomery[21], Graham Murphy[9], Maxwell Myers[21], Derek S. Nelson[21], Adriana Ocampo[3], Ryan Olds[5], John Y. Pelgrift[21], Trevor Perkins[5], Jon Pineau[29], Devin Poland[2], Vaishnavi Ramanan[21], Debi Rose[1], Eric Sahr[21], Owen Short[27], Ishita Solanki[1], Dale Stanbridge[21], Brian Sutter[5], Zachary Talpas[1], Howard Taylor[9], Bo Treiu[3], Nate Vermeer[5], Michael Vincent[1], Mike Wallace[30], Gerald Weigle[30], Daniel R. Wibben[21], Zach Wiens[1], John P. Wilson[9] & Yifan Zhao[4]

[1]Southwest Research Institute, Boulder, CO, USA. [2]NASA Goddard Space Flight Center, Greenbelt, MD, USA. [3]NASA Headquarters, Washington, DC, USA. [4]Arizona State University, Tempe, AZ, USA. [5]Lockheed Martin Space, Littleton, CO, USA. [6]Massachusetts Institute of Technology, Cambridge, MA, USA. [7]University of Central Florida, Orlando, FL, USA. [8]California Institute of Technology, Pasadena, CA, USA. [9]Johns Hopkins University Applied Physics Laboratory, Laurel, MD, USA. [10]Northern Arizona University, Flagstaff, AZ, USA. [11]Lowell Observatory, Flagstaff, AZ, USA. [12]Rheinisches Institut für Umweltforschung an der Universität zu Köln, Cologne, Germany. [13]University of Oxford, Oxford, UK. [14]NSF's National Optical-Infrared Astronomy Research Laboratory, Tucson, AZ, USA. [15]London Stereoscopic Company, London, UK. [16]CNRS, Observatoire de la Côte d'Azur, Laboratoire J.-L. Lagrange, Nice, France. [17]DLR Institute of Planetary Research, Berlin, Germany. [18]Muon Space, Mountain View, CA, USA. [19]University of Maryland, College Park, MD, USA. [20]American University, Washington, DC, USA. [21]KinetX Space Navigation and Flight Dynamics Practice, Simi Valley, CA, USA. [22]Indigo Information Services, Tucson, AZ, USA. [23]Red Canyon Software, Denver, CO, USA. [24]Cornell University, Ithaca, NY, USA. [25]Marshall Space Flight Center, Huntsville, AL, USA. [26]Lauffer Space Engineering, Littleton, CO, USA. [27]Teton Cyber Technology, Littleton, CO, USA. [28]Rochester Institute of Technology, Rochester, NY, USA. [29]Stellar Solutions, Denver, CO, USA. [30]Big Head Endian, Burden, KS, USA. ✉e-mail: hal.levison@swri.org

# Methods

## Observations

The analysis presented here is based on panchromatic (350–850 nm) images taken with Lucy's LOng Range Reconnaissance Imager, hereafter L'LORRI, which is a 20.8-cm, $f$/13 telescope feeding a 1,024 × 1,024-pixel CCD focal plane[35]. L'LORRI has a field of view of 0.29° and a pixel size of 5 µrad. It was primarily used in three distinct observation campaigns during the encounter. (1) Optical navigation reconstruction images were designed to precisely determine the trajectory of Lucy. They were taken daily during the period of ±4 days of encounter ($t_{CA}$ = −4 to +4 days) and every 15 min from $t_{CA}$ = −2 h to +2 h. (2) High-resolution close-approach images were taken every 15 s from $t_{CA}$ = −10 min to +9 min, then with 1-min cadence until +55 min. (3) Post-encounter light-curve photometry was acquired from $t_{CA}$ = +4 h to +95 h. Three exposures were taken at a cadence of 1 h. At this time, the Dinkinesh–Selam system was unresolved. To minimize data volume, these data were taken in L'LORRI's so-called 4 × 4 mode, which bins the data by 4 × 4 pixels during the CCD readout.

## Light-curve analysis

The orbital period of Selam and the rotational period of Dinkinesh can be determined using the post-encounter light-curve photometry described above in the 'Observations' section. Instrumental magnitudes of the system were extracted from the images using a 1.5-pixel-radius aperture. The small aperture served to exclude contamination from nearby stars. The formal errors from the extraction were scaled upward by a factor of 1.545 to adjust the reduced $\chi^2$ to be 1 before determining the final uncertainties on the fitted results. There were 267 images analysed.

The data were compensated for the changing distance as well as correcting to a constant solar phase angle using a phase coefficient of 0.06 mag per °. The phase angle varied from 60.52° at the start to 59.67° at the end. The observing direction changed little over the 3.5 days and these corrections remove these slight changes, leaving only a record of the global photometric properties of the system. The resulting light curve is shown in Extended Data Fig. 1 in units of relative flux.

We analysed the light curve with an iterative process designed to separate the contributions to the total flux from Dinkinesh and Selam. As the first step, a model was constructed that consisted of a Fourier series expansion of the light curve combined with a period for each object. The reference time for the rotational phase was arbitrarily set to the time of the first data point for both objects. The mean flux of Dinkinesh was a free parameter in the model. Also, we iteratively varied the Selam/Dinkinesh mean flux ratio. This ratio is constrained by the close-approach resolved images (Fig. 1d, for example), which show that the ratio of the visible areas of the two objects is 0.25. The two objects are also seen to have similar surface brightness, and so the unresolved flux ratio is also 0.25. This ratio was assumed to be at minimum light for both objects because Selam is viewed edge-on. An iterative correction was applied after separating the light curves to correct from the minimum to the mean flux and the final mean flux ratio was set at 0.33 (corresponding to a magnitude difference of 1.3).

The model parameters were determined in a series of iterative steps. The first pass fit set a reasonable mean flux for Dinkinesh and the Fourier terms were disabled. At this point, only Selam was free to be adjusted to fit the data. The data were scanned in period. At each step, a best-fit Fourier series was computed and the $\chi^2$ was recorded. The lowest $\chi^2$ period gave a preliminary value of 51.76 h for Selam. This model was subtracted from the light-curve data and a similar scan was performed on the Dinkinesh-only data. The Dinkinesh scan returned two interesting minima in $\chi^2$ at periods of roughly 3.7 and 4.3 h. Note that all periods assume that the light curve is double-peaked.

Given the two preliminary periods, the data were then fitted with the full model from the two objects and all free parameters were optimized simultaneously with an amoeba $\chi^2$ minimization (ref. 36 Chapter 10.4). Using the amoeba fit as the starting point with the a posteriori correction to the uncertainties, a second Markov chain Monte Carlo fit (see ref. 37) was run for the model. There were 18 data points that were excluded because of unreasonably large residuals (see the discussion below). The final fitted light curves revealed amplitudes of 0.82 mag for Selam and 0.25 mag for Dinkinesh.

The Selam rotation period was determined to be 52.44 ± 0.14 h from this fit, but it is also attributed to its orbital period about Dinkinesh because it is probably tidally locked, as shown by the presence of mutual events. The resulting phased light curves are shown in Fig. 2.

The variation in flux for the two objects coincidentally are about the same. Dinkinesh is much larger, which implies that it has a smaller relative variation in its flux. The light curve of Selam is well fit by two Fourier terms that capture the slightly asymmetric maximum and slightly broadened minima. The light curve of Dinkinesh is considerably more complicated; both the minima and maxima are asymmetric but there are also clearly higher-order variations seen. In this case, a four-term Fourier fit was required and even this does not fully capture all of the detail in the curve. For instance, one of the minima is sharper than can be followed with a four-term fit. The rotation period of Dinkinesh was determined to be 3.7387 ± 0.0013 h (the 4.3-h period discussed above was determined to be an alias).

The outliers that were flagged during the light-curve fitting, which are shown in red in the figures, are also of interest because they occur at a coherent rotation phase following a similar time after the two light-curve minima for Selam. A reasonable explanation for these low points is a mutual event between the two bodies. These could, in general, be from the bodies occulting each other from the perspective of the spacecraft or from casting shadows on one another. Fortunately, the timing of these minima allows us to determine which.

Looking at the photometry as a function of time, the low points appear at a regular interval at half the rotation period of Selam. Geometric constraints from the absolute timing indicate that the events are shadow transits of each other and not physical obscuration along the line of sight (occultations). Furthermore, the timing clearly indicates that the orbital motion of Selam is retrograde, as is true for the rotation of Dinkinesh as well. The first and third dips seen in time are inferior shadowing events, whereas the middle dip is a superior event. In the phased plot, the two inferior events overlay each other and trace out a more complete light curve of an event. The superior event has fewer measurements and shows an incomplete profile of the dip that misses the maximum eclipse point that must be in the middle between the two sets of points.

## Shape

The digital shape model used for this study (see Fig. 3) was generated by applying classical stereophotogrammetry techniques (ref. 38 and references therein) to L'LORRI imagery. A total of 48 images with a best ground-sampling distance ranging from about 10 m per pixel to 2.2 m per pixel were chosen from the high-resolution close-approach images described in the 'Observations' section. These were used to establish a network of 3,000 control points, which served as an input for the bundle adjustment process. Further, thanks to the very good noise and sensitivity performance of the L'LORRI imager, and to its comparatively large field of view, we could identify about 20 catalogue field stars in the Dinkinesh fields throughout the encounter. These star positions were used in the determination of the stereophotogrammetric adjustment, and contributed considerably to stabilize the solution.

As a result, the camera extrinsic matrices were determined, which describe the transformation between the camera's and the body-fixed reference system. These transformation matrices were then used to triangulate surface points from homologous image points, which were derived by means of dense stereo matching[39]. The resulting dense point cloud (about $5 \times 10^6$ 3D points) was then connected into a regular

triangular mesh. The shape model derived from stereo reconstruction has an estimated scale error of about 1.4% and covers about 45% of the body's surface. To produce a closed shape, and allow an estimation of the body volume, the unseen hemisphere has been approximated with an analytical solid figure. For this purpose, we chose a generalized super-ellipsoid[40], whose implicit representation is given by the function

$$1 = \left| \frac{x}{a} \right|^k + \left| \frac{y}{b} \right|^m + \left| \frac{z}{c} \right|^n$$

in which $x$, $y$ and $z$ are the standard Cartesian coordinates. A fit to the reconstructed hemisphere leads to $a = 0.40$, $b = 0.40$, $c = 0.35$ km, $k = m = 2$ and $n = 1.35$. The generalized super-ellipsoid provides a better match to the 'top' shape of Dinkinesh than a conventional triaxial ellipsoid.

We estimated the uncertainty in the volume of Dinkinesh from the difference between the shape model and the super-ellipsoid convex shell. For the hemisphere covered by imaging, the difference in volume is 4.7%. To be conservative, we round this and apply an arbitrary factor of two margin to arrive at the volume uncertainty of ±10%. This uncertainty is propagated to quantities derived from the volume. In particular, we note that the volume-equivalent radius of Dinkinesh is calculated as $r_{veq} = (3V/4\pi)^{1/3}$, rather than from direct distance measurements.

The dimensions of the two lobes of Selam were found by fitting ellipses to orthogonal axes in several resolved images of Selam from different viewing angles. The inner lobe of Selam is fit with an ellipsoid measuring 240 × 200 × 200 m. The outer lobe is measured at 280 × 220 × 210 m. Uncertainties were estimated to be 10% per axis by adjusting the ellipsoidal fits until they were visually too large or too small to match the images. Combining the above values, we calculate a total system volume of $V_{tot} = 2.06 \pm 0.20 \times 10^8$ m³.

## Mass and density

System density can be estimated from the orbital period and relative semimajor axis of the two bodies. As we describe in the main text, the centre-of-figure separation between Dinkinesh and Selam was 3.11 ± 0.05 km at the time of the fly-by. The eccentricity of Selam's orbit is not directly derivable from existing data, although it can be constrained. The regular phasing of the light-curve minima collected before encounter from the ground[14] and from Lucy (Fig. 2) is consistent with a near-circular orbit, given our inference (Fig. 2) that these minima are caused by mutual eclipses. We would expect the eccentricity of Selam to be near zero, given that tidal timescales for orbit circularization are on the order of $10^6$–$10^7$ years. The ages of asteroid pairs for which one of the members of the pair has subsequently undergone a mass-shedding event leading to the formation of a satellite suggest that binary-YORP effects[41] might shorten the circularization timescale to less than about $10^6$ years (refs. 16,42). Thus we assume $e = 0$ in the analysis performed here. Ground-based light-curve observations, taken at several epochs, can better constrain any orbital eccentricity that might exist.

Assuming that Selam is in a circular orbit about Dinkinesh and has an orbital period of 52.67 ± 0.04 h, we derive a system mass of $4.95 \pm 0.25 \times 10^{11}$ kg (GM = 33.0 ± 1.6 m³ s⁻²) from Kepler's third law. In the 'Shape' section, we calculate a total system volume of $V_{tot} = 2.06 \pm 0.20 \times 10^8$ m³. Combining the system mass and volume, we derive a bulk density of $\rho = 2,400 \pm 350$ kg m⁻³. We add the caveat that, if the assumption of zero eccentricity is incorrect and the separation observed at the time of the fly-by differs from the semimajor axis, it would introduce a systematic error into the calculation of density. Conversely, however, the range of likely density for an S-type asteroid, as discussed below, constrains the maximum eccentricity to be on the order of 0.1 and the assumption of zero eccentricity is fully consistent with known asteroid properties.

## Angular momentum

Knowledge of the component masses and the spin state can be combined to calculate the angular momentum of the system. For simplicity, we assume that the moment of inertia of Dinkinesh can be adequately represented by a sphere of volume-equivalent radius. Assuming that Selam is tidally locked, the contribution to the angular momentum from its spin is small. Likewise, the orbital motion of Dinkinesh around the barycentre is small and we ignore it. The system angular momentum is nearly equally divided between the spin of Dinkinesh, $L_{spin} = 11.2 \pm 1.9 \times 10^{12}$ kg m² s⁻¹, and the orbital motion of Selam, $L_{orb} = 8.0 \pm 4.0 \times 10^{12}$ kg m² s⁻¹. The total angular momentum of the system is $L_{sys} = 19.3 \pm 4.4 \times 10^{12}$ kg m² s⁻¹. The normalized angular momentum, $\alpha_L$, is computed from the total system angular momentum divided by the angular momentum of a sphere containing the total mass of the system rotating at the maximum rate for a cohesionless rubble pile[43]. That rate is given by $\omega_{max} = (4\pi\rho G/3)^{1/2}$, corresponding to a spin period of $T_{max} = 2.13$ h, that is, the observed main-belt spin barrier. We find $\alpha_L = 0.88$, consistent with that expected for a binary produced by fission[26].

35. Weaver, H. A. et al. The Lucy Long Range Reconnaissance Imager (L'LORRI). *Space Sci. Rev.* **219**, 82 (2023).
36. Press, W. H., Teukolsky, S. A., Vetterling, W. T. & Flannery, B. P. *Numerical Recipes in C. The Art of Scientific Computing* 2nd edn (Cambridge Univ. Press, 1992).
37. Foreman-Mackey, D., Hogg, D. W., Lang, D. & Goodman, J. emcee: the MCMC hammer. *Publ. Astron. Soc. Pac.* **125**, 306 (2013).
38. Preusker, F. et al. The global meter-level shape model of comet 67P/Churyumov-Gerasimenko. *Astron. Astrophys.* **607**, L1 (2017).
39. Wewel, F. Determination of conjugate points of stereoscopic three line scanner data of Mars 96 mission. *Int. Arch. Photogram. Remote Sensing* **31**, 936–939 (1996).
40. Ni, B., Elishakoff, I., Jiang, C., Fu, C. & Han, X. Generalization of the super ellipsoid concept and its application in mechanics. *Appl. Math. Model.* **40**, 9427–9444 (2016).
41. Ćuk, M. & Burns, J. A. Effects of thermal radiation on the dynamics of binary NEAs. *Icarus* **176**, 418–431 (2005).
42. Pravec, P. et al. Asteroid pairs: a complex picture. *Icarus* **333**, 429–463 (2019).
43. Pravec, P. & Harris, A. W. Fast and slow rotation of asteroids. *Icarus* **148**, 12–20 (2000).

**Acknowledgements** The Lucy mission is financed through the NASA Discovery programme on contract no. NNM16AA08C. We thank the entire Lucy mission team for their hard work and dedication.

**Additional information**
**Correspondence and requests for materials** should be addressed to Harold F. Levison.

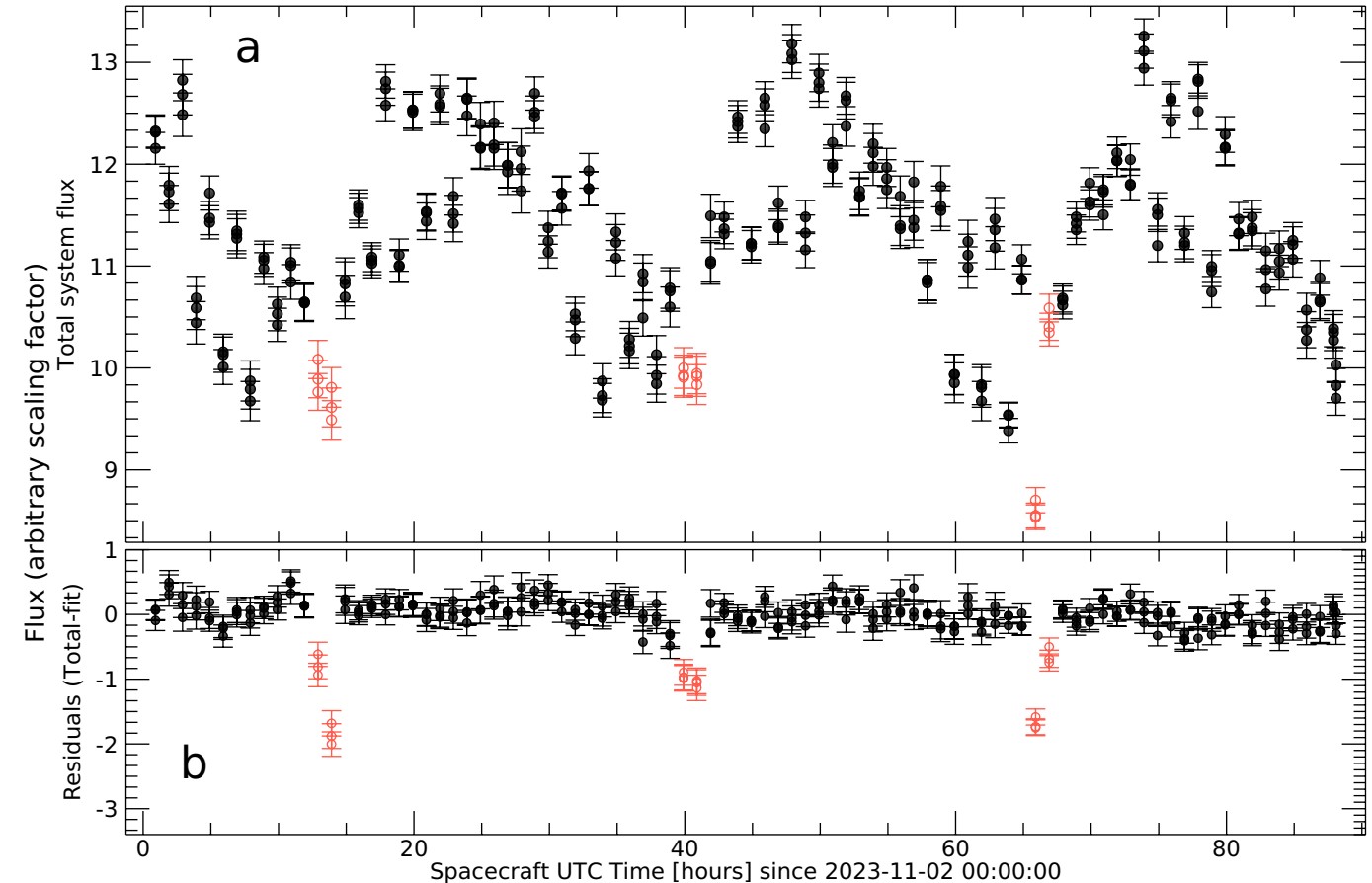

**Extended Data Fig. 1 | Post-encounter photometry of the Dinkinesh–Selam system. a**, The observed flux (with arbitrary scale) of the system as a function of time. **b**, The residuals to the fit described in the 'Light-curve analysis' section in Methods. The solid black points are the data used in the combined light-curve fit. The hollow red points are those that were excluded from the fit owing to being affected by mutual events between the components.