## [Peer Review File · Nature]

Manuscript Title: A Contact Binary Satellite of the Asteroid (152830) Dinkinesh

Reviewer Comments & Author Rebuttals

Reviewer Reports on the Initial Version:

Referees' comments:

Referee #1 (Remarks to the Author):

Review of

The Discovery of a Contact-Binary Satellite of the Asteroid (1523830) Dinkinesh by the Lucy Mission

This work discusses the flyby of asteroid Dinkinesh and the discovery that it is a binary asteroid with a dual lobed secondary during the Lucy mission. It describes the physical characteristics of the two objects as well as discusses possibilities for its formation.

I enjoyed reading this work and found it to be easy to read and understand. I suggest some clarifications that may help the readers understand the material better.

This work is quite relevant because the Lucy mission is active and is expect to produce large amounts of data. Additionally, this work helps demonstrate the capabilities of the instrument platform and the type of analysis that can be done during the Lucy mission. This work will be of interest to the readers of Nature because of discovery of a contact secondary for a previously unknown binary.

Main comment

The paper fully commits to YORP as not only the main formation mechanism, but the only one. While the observed characteristics are inline with YORP, the paper does not provide evidence of other cause elimination, i.e. various impact scenarios. If this cause can be eliminated, then it should be discussed as to why it is not possible.

Additionally, YORP is impacted by the physical shape of surface features on an asteroid. Because the backside of Dinkinesh was not observed, I do not understand how YORP can be calculated or constrained. YORP could either be spinning up or down.

Minor comments

Figure 1 - Subimage D. From what is written, I believe that you constructed what Dinkinesh looked like before a hypothesized collapse happen that formed the trough. This is not clear from the text.

- The image isn't reconstructed, but it is a simulation of what it might have looked like.

- How far was the two components moved (i.e. how much overlap) was needed to form the picture (in meters).

Figure 1 -

According to IAU, coordinates for asteroids are positive and negative latitude, rather than north and south. If you use north for the pole, then it would be good to denote it, especially because Dinkinesh's orientation is retrograde.

Figure 1 - blue/green arrows. The text says that they are at "the northern boundary of the ridge". Does that mean the south pole is at the top of the image?

Line 121 "but are of order", should be "and are of order..."

Lines 122-124. The authors discuss a mechanism that can shorten the timescale for circularization to less than 10^6 yrs. Why is that an important consideration. I did not notice any limitations as to time constrains, especially considering YORP effects can vary in strength, and thus, time.

Line 142-143. "nearly ubiquitous among small binary asteroids". Can you provide the actual number and what fraction that constitutes?

Line 179. "Fig. 1 shows" — seems too strong with insufficient evidence. Maybe use "suggests"

Lines 194-201. The constrains that Selam places seems overstated. First, just because the lobes have the same size does not require a specific mechanism that drives that behavior. Simulations might provide some insight as to how this could happen, although that would be out of scope of this paper.

Second, there is insufficient evidence about what is occurring in the shadow to state that they are distinct bodies. While two objects is within the range of possibilities, having the connection hidden by shadow is a viable possibility also.

Figure 3. I like this figure, which has a lot of information packed into something so simple.

- I suggest that the orientation of the center (top) shape model be the same as Figure 1. It appears that it was flipped (the lobe is on the right in figure 1 and the left in figure 3).

- The green and blue large dots are pretty hard to notice.

- Can you denote on an image where the physical measurements of the ridge is taken. I was unable to correlate the texts' description to the images (150x x 40m and 230 x 100m)

- Is Z positive latitude?

Figure 4. Should collision and YORP spin up be included? If not, why?

Line 231. "... resolved images the area ratio is 0.25". I did not follow what the ratio was of. The term

“area ratio” seems insufficient to describe what components are being measured.

Line 262. “These could, in general, be ...” It seems that we have sufficient information to model if these effects came from either occulting or casting shadows. This sentence seems to suggest that we do not know which. However, the following paragraph talks about shadows. Please clarify.

Line 271 and following. Shape.

- The bundle adjustment allows for the correction of spacecraft geometry on the stereo processing. How much did the spacecraft position and pointing change? How much did it improve the solution?

- Shape modeling over a single pass has a difficult time establishing the absolute size of the object. There is aliasing between the distance of the object and size. Because we did not orbit the object, the distance to the object is one of the weakest components of our trajectory. How was the size controlled for, especially in light of a bundle adjustment?

Line 272 — What is the justification for a “factor of two margin” for the error estimate?

Line 282 — What is the justification for a 10% per axis” for the uncertainties?

Eric E Palmer

Referee #2 (Remarks to the Author):

A. Summary of the key result:

This manuscript presents the discovery of a contact binary satellite around asteroid Dinkinesh, based on flyby observations by NASA's Lucy mission. The authors provide essential details about the newly discovered binary satellite, including its size, mass, density, orbital properties, shape, and geomorphological characteristics. Additionally, the authors propose that the satellite may have formed from a disk generated by mass shed from the primary asteroid. I extend my congratulations to the Lucy team for this remarkable discovery during their flyby observations. Such a significant achievement would not have been possible without the team's dedicated efforts during the observation campaign.

B, Originality and significance:

This study represents the first discovery of a contact binary satellite around an asteroid, demonstrating its novelty in the field. The absence of prior publications on this discovery reinforces its significance. While there may be theoretical papers in recent literature citing a general news on Dinkinesh's satellite, these works do not diminish the novelty of this study; rather, they underscore the importance of this new finding. Furthermore, the presence of a contact binary satellite orbiting an asteroid with an equatorial ridge and trough holds profound implications for small-body formation, enhancing the broader significance of this discovery. Given the uniqueness and potential impact of this finding, I believe it warrants publication in a prestigious journal like Nature, following rigorous peer review.

C. Data & methodology: validity of approach, quality of data, quality of presentation.

The primary data utilized in this study include images in Fig. 1 and light curve data in Fig. 2, both of which are of exceptional value and quality. Given the constraints of the flyby mission on data acquisition conditions and volume, the formal quality and quantity of the data may be limited. However, the information contained within the data is immensely valuable. Furthermore, the authors appropriately acknowledge and discuss the limitations of the data.

Major Comment: While the methodology and argumentation based on the data are generally sound, this manuscript would benefit from a more thorough discussion of the observations before publication in Nature. Specifically, there is a lack of discussion regarding the geomorphological evidence that supports the authors' interpretations and conclusions. While I generally agree with the authors' interpretations, such as the contact binary nature of the satellite of Dinkinesh, there is insufficient discussion of the observations that form the basis of these interpretations. Although the photographs of Dinkinesh are great, the authors do not adequately discuss what is observed in the images and why they interpret the satellite as a contact binary. For instance, the contact point between the two lobes of the satellite is not visible in any of the images, as indicated by the authors in Line 199. Therefore, it is essential to elucidate the reasoning behind the interpretation of the contact binary nature. Depending on the nature of the supporting evidence, the confidence level in the interpretation of a contact binary may vary.

I think a contact binary requires two things: (1) the lobes are distinct bodies, and (2) the lobes are in contact. As for (1), the authors clearly state that they are distinct bodies in lines 199-200. But the reason for this statement is not provided. Please indicate the reason why the authors believe that two lobes are distinct bodies despite the lack of image showing the contact point. I do not think this evidence have to be very strong, but there has to be a reason.

The statement (2) may be a bit easier to support, but I cannot find the clear discussion on the reason for this either. One thing I suspect is that their mutual rotation around each other, estimated from the tidal lock configuration, is too slow to be separated from each other. This may be obvious to the authors, but it is not obvious to me. The authors may have something else as their reasoning. Please state this.

Having said that, I think this manuscript deserves publication in nature regardless of the degree of confidence and the nature of the observation. Nevertheless, I believe that the presentation of such evidence and discussion is fundamentally impotent to warrant the integrity of this manuscript and the journal. A minor comment: Regarding the color scheme for the arrows in Figure 2, using more distinctively different colors would indeed improve clarity and readability for readers. The authors may consider selecting colors with higher contrast or using a color palette that ensures clear differentiation between the arrows.

D. Appropriate use of statistics and treatment of uncertainties

The authors present many important orbital and geophysical properties of the primary asteroid and satellite. But some observed values could have error bars as listed as follows.

Lines 107-109: For the periods of Salem's orbital period, even if the authors cannot provide precise error bars due to limited data or other constraints, they should acknowledge this limitation in the text. They can briefly explain why it's challenging to estimate the error bars and discuss the

implications of this uncertainty for their analysis.

Lines 113-114: Regarding the alignment of Dinkinesh and the two lobes of Selam, providing an estimate of the angle of misalignment or the degree of uncertainty would enhance the discussion. While obtaining a precise value may be challenging due to the limited rotation of Selam during the flyby, the authors can still discuss the general alignment and any potential deviations. They could mention the methods used to determine the alignment and discuss the limitations or uncertainties associated with these measurements. This would provide readers with a better understanding of the reliability of the alignment assessment.

Lines 136-138: For the statement regarding Dinkinesh's obliquity, providing an error estimate would indeed enhance the discussion. If the authors have performed any uncertainty analysis or sensitivity tests to determine the reliability of the obliquity estimate, they should mention this in the manuscript. If not, they can acknowledge the lack of a precise error estimate and discuss any potential sources of uncertainty that could affect the determination of Dinkinesh's obliquity.

Lines 141-142: Similarly, for the statement about the alignment of Dinkinesh's heliocentric orbit, Selam's orbit, and Dinkinesh's equatorial plane, providing an estimate of the uncertainty would improve clarity. The authors could discuss the factors influencing the alignment and any limitations in the measurements or calculations that could introduce uncertainty. Even if a precise error estimate is challenging to obtain, acknowledging the potential range of uncertainty would help readers better understand the reliability of the alignment assessment.

E. Conclusions: robustness, validity, reliability

Line 71-72: Providing the rationale behind the inference about Dinkinesh's ridge and trough would indeed enhance clarity. The suggested revision provides a clearer structure for the sentence and prompts the authors to explicitly mention the basis for their inference. It could be rewritten in the form like “[Geomorphologic observations? A simple mechanical calculation?] lead us to infer that Dinkinesh's ridge and trough are likely the result of mass failure resulting from YORP torques, and the reaccretion of material”.

Line 166-167: L-chondrite meteorites

L-chondrite is one of the meteorite clans for S-type asteroids. But it is not the only clan; there are also LL and H types. For example, asteroid Itokawa turned out to be LL chondrites (Tsuchiyama et al. 2011 Science). Fortunately, LL, L, and H chondrites are too different in density (~ 10%). The authors have conscientiously included about 10% of error in their macro porosity estimate (20 – 30 %). However, the logic to reach this porosity estimate is a bit rough. Please discuss the source of error in density estimate here. Also, if you have a reflectance spectrum of Dinkinesh, it could help identify which type of ordinary chondrite the author should use as an analogue material.

Line 179: “Fig.1 shows that Dinkinesh suffered a global structural failure in its past”.

This is a very interesting and important statement. However, it is an interpretation, not observation. What is the observation that leads the authors to this interpretation? Please discuss the key observation that the authors have found in the images in Fig. 1.

Line 194-195 & Fig. 4

Fig. 4 illustrate that Selam may have formed from a ring of small fragments shed off from Dinkinesh surface, suggesting that Selam would be rubble pile. The manuscript, however, does not discuss about the observations for or against rubble pile nature of Selam. the authors need to discuss what kind of information they have gathered so far from the flyby observations.

Examples are a possible equatorial ridge and craters on Selam. Do their geomorphological properties support rubble pile nature? The authors also point out that Selam may have significant internal strength (Line 151-152). Does this argue against rubble pile nature?

Such a discussion will clarify the degree of confidence for the hypothesis illustrated in Fig. 4. Also, I believe that an important role of an initial report for mission results is to present key observation results more than a hypothesis, although an attractive hypothesis is useful.

Having said that, if Lucy's team has not been able to reach conclusion about the rubble pile nature of Selam, I do not think that the authors have to conduct further analysis. Nevertheless, if this is the case, the authors should state that observations obtained so far does not support the rubble pile nature of Selam yet.

Can a contact binary lobe be a rubble pile and still support their own distinct shapes without collapsing into one lobe due to their self-gravity? In fact, authors state that these observations support internal strengths of Selam (Line 15-141). However, recent studies on Ryugu and Bennu surfaces based on artificial disturbances show that their cohesion is extremely small (<1.3 Pa for Ryugu Arakawa et al., 2020 Science, <0.001 Pa for Bennu Lauretta et al. 2022 Science). How can these two notions be reconciled? I do not think that this problem does not have to be resolved in this manuscript. But at least this apparent contradiction should be mentioned as an open question.

F. Suggested improvements: experiments, data for possible revision

I do not think substantial addition of experiments or data are necessary.

I believe that revision in only writing should be sufficient.

Line 203. Inference for small bilobed bodies like Itokawa. This is a very interesting inference. I would suggest citing [28] here as well.

G. References: appropriate credit to previous work?

Some citations to references are suggested to add, but references are generally very appropriate.

H. Clarity and context: lucidity of abstract/summary, appropriateness of abstract, introduction and conclusions

I think abstract, introduction, and conclusions are written well. In particular, the abstract presents fundamental properties of Dinkinesh, such as diameter and the presence of satellite, and equatorial ridge. This is very good.

Author Rebuttals to Initial Comments:

We are returning the manuscript entitled "A Contact Binary Satellite of the Asteroid (152830) Dinkinesh" for your continued consideration for publication in Nature. As we describe in more detail below, we have addressed the referee comments. In addition, we have modified the text to fit into Nature's length and format requirements. In particular, we have moved the discussion of the system mass and density to a new section in Methods. We also slightly broadened our interpretation of Selam's tilted ridge (lines 164-165), and corrected a mistake on the cadence of observations (lines 289 and 290).

We now directly respond to the referees' comments. Note that the lines numbers we give are for the current manuscript.

> Referee #1 (Remarks to the Author):

>

> Review of

>

> The Discovery of a Contact-Binary Satellite of the Asteroid (1523830)
> Dinkinesh by the Lucy Mission

>

> This work discusses the flyby of asteroid Dinkinesh and the discovery
> that

> it is a binary asteroid with a dual lobed secondary during the Lucy
> mission. It describes the physical characteristics of the two objects as
> well as discusses possibilities for its formation.

>

> I enjoyed reading this work and found it to be easy to read and
> understand.

> I suggest some clarifications that may help the readers understand the
> material better.

>

> This work is quite relevant because the Lucy mission is active and is
> expect

> to produce large amounts of data. Additionally, this work helps
> demonstrate the capabilities of the instrument platform and the type of
> analysis that can be done during the Lucy mission. This work will be of
> interest to the readers of Nature because of discovery of a contact
> secondary for a previously unknown binary.

>

>

> Main comment

>

> The paper fully commits to YORP as not only the main formation mechanism,
> but the only one. While the observed characteristics are inline with
> YORP,

> the paper does not provide evidence of other cause elimination, i.e.
> various impact scenarios. If this cause can be eliminated, then it
> should

> be discussed as to why it is not possible.

>

> Additionally, YORP is impacted by the physical shape of surface features

on
> an asteroid. Because the backside of Dinkinesh was not observed, I do
not
> understand how YORP can be calculated or constrained. YORP could either
be
> spinning up or down.

While it is true that we discuss the effects of YORP (or now called
'radiation effects' someplaces) on the current system at several locations
in the main text (lines 119, 156-157, 203. and 402), we only invoke it as a
formation mechanism near the end of the main text (line 188) and in Fig.4.
There is no getting around the fact that YORP is important to the current
dynamical evolution of the system, and so we made no modifications there.
The collisional origin of top-shaped asteroids is now addressed in the
caption of Fig.4 and we point to this text at line 188-189.

>
> Minor comments
>
> Figure 1 - Subimage D. From what is written, I believe that you
constructed
> what Dinkinesh looked like before a hypothesized collapse happen that
formed
> the trough. This is not clear from the text. - The image isn't
> reconstructed, but it is a simulation of what it might have looked like.
-
> How far was the two components moved (i.e. how much overlap) was needed
to
> form the picture (in meters).

These changes were made and the information added to the caption to Fig. 1.

> Figure 1 -
> According to IAU, coordinates for asteroids are positive and negative
> latitude, rather than north and south. If you use north for the pole,
then
> it would be good to denote it, especially because Dinkinesh's
> orientation is retrograde.
>
> Figure 1 - blue/green arrows. The text says that they are at "the
> northern boundary of the ridge". Does that mean the south pole is
> at the top of the image?

We now discuss the orientation of the images in the caption.

> Line 121 "but are of order", should be "and are of
> order"

This text has been deleted in response to the next comment.

>
> Lines 122-124. The authors discuss a mechanism that can shorten the
> timescale for circularization to less than 10^6 yrs. Why is that an
> important consideration. I did not notice any limitations as to time
> constrains, especially considering YORP effects can vary in strength, and
> thus, time.

Our intent is to justify our assumption that $e=0$. We have made it clearer.
(Line 398-399)

> Line 142-143. "nearly ubiquitous among small binary
> asteroids". Can you provide the actual number and what fraction that
> constitutes?

Well, it is complicated and we don't really have room to discuss. Thus,
we have added a reference on line 157 where these numbers can be found.

> Line 179. "Fig. 1 shows" - seems too strong with
> insufficient evidence. Maybe use "suggests"

We used 'strongly suggests' (line 187)

> Lines 194-201. The constrains that Selam places seems overstated.
First,
> just because the lobes have the same size does not require a specific
> mechanism that drives that behavior. Simulations might provide some
> insight as to how this could happen, although that would be out of scope
of
> this paper.

We weakened the statement by replacing 'implies' to 'argues' and 'must
favor' to 'favors' (lines 207 & 208)

> Second, there is insufficient evidence about what is occurring in the
shadow
> to state that they are distinct bodies. While two objects is within the
> range of possibilities, having the connection hidden by shadow is a
viable
> possibility also.

We agree that we did not explain this very well. There is evidence that
the neck is significantly smaller than the diameters of the lobes, which
justifies the conclusion that Selam is a contact binary. This discussion
now occurs in the paragraph starting on line 131. We also modified the
text on line 210. Note that the second referee also makes this point.

> Figure 3. I like this figure, which has a lot of information packed into
> something so simple.
>
> - I suggest that the orientation of the center (top) shape model be the
> same as Figure 1. It appears that it was flipped (the lobe is on the
right
> in figure 1 and the left in figure 3).

We would rather not. It makes more sense to us to show real images in a
frame where ecliptic north is up, but the models in a frames where the
rotational axis is up, because the obliquity of Dinkinesh is not 180 deg and
the spacecraft did not fly along the ecliptic. We have made this clearer
in the figure captions.

>

> - The green and blue large dots are pretty hard to notice.

We have made them bigger.

>

> - Can you denote on an image where the physical measurements of the ridge

> is taken. I was unable to correlate the texts' description to the
> images (150x x 40m and 230 x 100m)

We have added this information to Figure 3.

> - Is Z positive latitude?

Yes, we think that it is clear from the location A1, A2, and A3. We made no changes in response to this comment.

> Figure 4. Should collision and YORP spin up be included? If not, why?

No, for reasons now discussed in the figure caption.

> Line 231. "... resolved images the area ratio is 0.25".

> I did not follow what the ratio was of. The term "area
> ratio" seems insufficient to describe what components are being
> measured.

We clarified this discussion on Lines 310-315.

>

> Line 262. "These could, in general, be ..." It seems

> that we have sufficient information to model if these effects came from
> either occulting or casting shadows. This sentence seems to suggest that
> we do not know which. However, the following paragraph talks about
> shadows. Please clarify.

This is fixed (lines 347 & 348).

> Line 271 and following. Shape.

> - The bundle adjustment allows for the correction of spacecraft geometry
> on

> the stereo processing. How much did the spacecraft position and pointing
> change? How much did it improve the solution?

>

> - Shape modeling over a single pass has a difficult time establishing
> the

> absolute size of the object. There is aliasing between the distance of
> the

> object and size. Because we did not orbit the object, the distance to
> the

> object is one of the weakest components of our trajectory. How was the
> size controlled for, especially in light of a bundle adjustment?

Those two questions are related. The reviewer is correct in saying that

the retrieval of shape, pose and relative position is a challenging problem in stereo photogrammetry, especially when performed from a fly-by geometry. In the case of the Dinkinesh fly-by, thanks to the very good noise and sensitivity

performance of the LLORRI imager, and to its comparatively large FOV, we could identify catalog field stars in the Dinkinesh fields throughout the encounter. This was key for the determination of a reliable model. We used a total of about 20 stars, some of which were present in consecutive fields. The presence of those stars allowed us to compute very accurate pointing angles in the plane of the image for those frames where only a single star was present. For those fields where two or more stars were present, we could accurately determine all of the three pointing angles. Fixing the pointing for those key frames greatly stabilized

the solution during the bundle adjustment. The corresponding adjustment of the relative position of the spacecraft/target distance was of the order of 300m. We estimate the final uncertainty on the object scale to be of the order of 1.4%. We now address this on lines 364 - 367 and 372 - 373.

> Line 272 - What is the justification for a "factor of two
> margin" for the error estimate?

This was a conservative measure, which is now explained on line 382.

> Line 282 - What is the justification for a 10% per axis" for
> the uncertainties?

The uncertainties were estimated by eye. This is now explained on line 387-390.

>
> Eric E Palmer

>
>
> Referee #2 (Remarks to the Author):

>
> A. Summary of the key result:
> This manuscript presents the discovery of a contact binary satellite
> around
> asteroid Dinkinesh, based on flyby observations by NASA's Lucy
> mission. The authors provide essential details about the newly discovered
> binary satellite, including its size, mass, density, orbital properties,
> shape, and geomorphological characteristics. Additionally, the authors
> propose that the satellite may have formed from a disk generated by mass
> shed from the primary asteroid. I extend my congratulations to the Lucy
> team for this remarkable discovery during their flyby observations. Such
> a
> significant achievement would not have been possible without the
> team's dedicated efforts during the observation campaign.

>
> B, Originality and significance:
> This study represents the first discovery of a contact binary satellite
> around an asteroid, demonstrating its novelty in the field. The absence
> of
> prior publications on this discovery reinforces its significance. While

> there may be theoretical papers in recent literature citing a general news

> on Dinkinesh's satellite, these works do not diminish the novelty of this study; rather, they underscore the importance of this new finding. Furthermore, the presence of a contact binary satellite orbiting an asteroid with an equatorial ridge and trough holds profound implications for small-body formation, enhancing the broader significance of this discovery. Given the uniqueness and potential impact of this finding, I believe it warrants publication in a prestigious journal like Nature, following rigorous peer review.

>

> C. Data & methodology: validity of approach, quality of data, quality of presentation. The primary data utilized in this study include images in Fig. 1 and light curve data in Fig. 2, both of which are of exceptional value and quality. Given the constraints of the flyby mission on data acquisition conditions and volume, the formal quality and quantity of the data may be limited. However, the information contained within the data is immensely valuable. Furthermore, the authors appropriately acknowledge and discuss the limitations of the data. Major Comment: While the methodology and argumentation based on the data are generally sound, this manuscript would benefit from a more thorough discussion of the observations before publication in Nature. Specifically, there is a lack of discussion regarding the geomorphological evidence that supports the authors' interpretations and conclusions. While I generally agree with the authors' interpretations, such as the contact binary nature of the satellite of Dinkinesh, there is insufficient discussion of the observations that form the basis of these interpretations. Although the photographs of Dinkinesh are great, the authors do not adequately discuss what is observed in the images and why they interpret the satellite as a contact binary. For instance, the contact point between the two lobes of the satellite is not visible in any of the images, as indicated by the authors in Line 199. Therefore, it is essential to elucidate the reasoning behind the interpretation of the contact binary nature. Depending on the nature of the supporting evidence, the confidence level in the interpretation of a contact binary may vary. I think a contact binary requires two things: (1) the lobes are distinct bodies, and (2) the lobes are in contact. As for (1), the authors clearly state that they are distinct bodies in lines 199-200. But the reason for this statement is not provided. Please indicate the reason why the authors believe that two lobes are distinct bodies despite the lack of image showing the contact point. I do not think this evidence have to be very strong, but there has to be a reason.

The first referee also made this point. We agree that we did not explain this very well. There is evidence that the neck is significantly smaller than the diameter of the lobes, which justifies the conclusion that Selam is a contact binary. This discussion now occurs in the paragraph starting on line 131. We also modified the text on line 210.

> The statement (2) may be a bit easier to support, but I cannot find the
> clear discussion on the reason for this either. One thing I suspect is
that
> their mutual rotation around each other, estimated from the tidal lock
> configuration, is too slow to be separated from each other. This may be
> obvious to the authors, but it is not obvious to me. The authors may have
> something else as their reasoning. Please state this.

We agree that we should have presented this argument. It too can now be
found in the paragraph starting on line 131.

> Having said that, I
> think this manuscript deserves publication in nature regardless of the
> degree of confidence and the nature of the observation. Nevertheless, I
> believe that the presentation of such evidence and discussion is
> fundamentally impotent to warrant the integrity of this manuscript and
the
> journal. A minor comment: Regarding the color scheme for the arrows in
> Figure 2, using more distinctively different colors would indeed improve
> clarity and readability for readers. The authors may consider selecting
> colors with higher contrast or using a color palette that ensures clear
> differentiation between the arrows.

We have changed the color of the of the purple arrows to orange.

> D. Appropriate use of statistics and treatment of uncertainties
> The authors present many important orbital and geophysical properties of
the
> primary asteroid and satellite. But some observed values could have error
> bars as listed as follows.
>
> Lines 107-109: For the periods of Salem's orbital period, even if the
> authors cannot provide precise error bars due to limited data or other
> constraints, they should acknowledge this limitation in the text. They
can
> briefly explain why it's challenging to estimate the error bars and
> discuss the implications of this uncertainty for their analysis.

We are confused by this comment because errors are provided on line 111 and
112.

> Lines 113-114: Regarding the alignment of Dinkinesh and the two lobes of
> Selam , providing an estimate of the angle of misalignment or the degree
of
> uncertainty would enhance the discussion. While obtaining a precise value
> may be challenging due to the limited rotation of Selam during the flyby,
> the authors can still discuss the general alignment and any potential
> deviations. They could mention the methods used to determine the
alignment
> and discuss the limitations or uncertainties associated with these
> measurements. This would provide readers with a better understanding of
the
> reliability of the alignment assessment.

We would need a shape model for Selam to quantify the alignment, which does

not exists yet. We have added the phrase 'appears to' to line 121 to emphasize the fact that the alignment is uncertain.

> Lines 136-138: For the statement regarding Dinkinesh's obliquity,
> providing an error estimate would indeed enhance the discussion. If the
> authors have performed any uncertainty analysis or sensitivity tests to
> determine the reliability of the obliquity estimate, they should mention
> this in the manuscript. If not, they can acknowledge the lack of a
precise
> error estimate and discuss any potential sources of uncertainty that
could
> affect the determination of Dinkinesh's obliquity.

During the reviewing phase of the manuscript, a new iteration for the shape model has been performed, which has also provided improved results for the spin axis orientation. The text has been accordingly updated, and includes now includes the uncertainties on line 150-151.

>
> Lines 141-142: Similarly, for the statement about the alignment of
> Dinkinesh's heliocentric orbit, Selam's orbit, and
> Dinkinesh's equatorial plane, providing an estimate of the uncertainty
> would improve clarity. The authors could discuss the factors influencing
> the alignment and any limitations in the measurements or calculations
that
> could introduce uncertainty. Even if a precise error estimate is
> challenging to obtain, acknowledging the potential range of uncertainty
> would help readers better understand the reliability of the alignment
> assessment.

This statement is a conclusion based on previous information where we presented our uncertainties as far as we have been able to estimate them. The statement starts with "It is therefore likely.." with already implies that there are uncertainties involved. Thus, we do not feel that any changes are necessary.

> E. Conclusions: robustness, validity, reliability
> Line 71-72: Providing the rationale behind the inference about
> Dinkinesh's ridge and trough would indeed enhance clarity. The
> suggested revision provides a clearer structure for the sentence and
> prompts the authors to explicitly mention the basis for their inference.
It
> could be rewritten in the form like "[Geomorphologic observations? A
> simple mechanical calculation?] lead us to infer that Dinkinesh's
> ridge and trough are likely the result of mass failure resulting from
YORP
> torques, and the reaccumulation of material".

We have add the phrase "Geomorphologic observations lead us to infer that" on line 72-73.

> Line 166-167: L-chondrite meteorites
> L-chondrite is one of the meteorite clans for S-type asteroids. But it is
> not the only clan; there are also LL and H types. For example, asteroid

> Itokawa turned out to be LL chondrites (Tsuchiyama et al. 2011 Science).
> Fortunately, LL, L, and H chondrites are too different in density (~10%).
> The authors have conscientiously included about 10% of error in their macro
> porosity estimate (20 - 30 %). However, the logic to reach this
> porosity estimate is a bit rough. Please discuss the source of error in
> density estimate here.

We have changed the text to explicitly state that we are using L's as an analog for the range of OC densities and changed the macroporosity numbers to align with the uncertainties in the bulk density number from the previous paragraph. (lines 173-175)

> Also, if you have a reflectance spectrum of
> Dinkinesh, it could help identify which type of ordinary chondrite the
> author should use as an analogue material.

Our IR spectra are not yet available.

> Line 179: "Fig.1 shows
> that Dinkinesh suffered a global structural failure in its past".
> This is a very interesting and important statement. However, it is an
> interpretation, not observation. What is the observation that leads the
> authors to this interpretation? Please discuss the key observation that
the
> authors have found in the images in Fig. 1.

This is already discussed in the caption for Fig. 1, which says "D) A simulated image of Dinkinesh with the trough removed. This is a modified version of the right panel in (A). We take the fact that the limb profile of Dinkinesh is smooth near the color transitions of this reconstruction to suggest that the trough is a result of a structural failure that moved the cyan region away from the remainder of the body." No changes were made in response to this comment.

> Line 194-195 & Fig. 4
> Fig. 4 illustrate that Selam may have formed from a ring of small
fragments
> shed off from Dinkinesh surface, suggesting that Selam would be rubble
> pile. The manuscript, however, does not discuss about the observations for
> or against rubble pile nature of Selam. the authors need to discuss what
> kind of information they have gathered so far from the flyby
observations.
> Examples are a possible equatorial ridge and craters on Selam. Do their
> geomorphological properties support rubble pile nature? The authors also
> point out that Selam may have significant internal strength (Line 151-152).
> Does this argue against rubble pile nature? Such a discussion will
clarify
> the degree of confidence for the hypothesis illustrated in Fig. 4. Also,
I
> believe that an important role of an initial report for mission results
is
> to present key observation results more than a hypothesis, although an
> attractive hypothesis is useful. Having said that, if Lucy's team

> has not been able to reach conclusion about the rubble pile nature of
> Selam, I do not think that the authors have to conduct further analysis.
> Nevertheless, if this is the case, the authors should state that
> observations obtained so far does not support the rubble pile nature of
> Selam yet.

A full discussion of this beyond the scope of this paper. Selam shows some evidence that it is a rubble pile, at least partially. We now point this out on lines 165-166.

> Can a contact binary lobe be a rubble pile and still support their own
> distinct shapes without collapsing into one lobe due to their self-
gravity?
> In fact, authors state that these observations support internal strengths
> of Selam (Line 15-141). However, recent studies on Ryugu and Bennu
surfaces
> based on artificial disturbances show that their cohesion is extremely
> small (<1.3 Pa for Ryugu Arakawa et al., 2020 Science, <0.001 Pa for
> Bennu Lauretta et al. 2022 Science). How can these two notions be
> reconciled? I do not think that this problem does not have to be resolved
> in this manuscript. But at least this apparent contradiction should be
> mentioned as an open question.

We agree that this is a key question. As we say in several places in the text, primitive asteroids like Ryugu and Bennu are clearly different from S-types like Dinkinesh. A further discussion of this issue is beyond the scope of this paper.

> F. Suggested improvements: experiments, data for possible revision
> I do not think substantial addition of experiments or data are necessary.
> I believe that revision in only writing should be sufficient.
>
> Line 203. Inference for small bilobed bodies like Itokawa. This is a very
> interesting inference. I would suggest citing [28] here as well.

Done (Line 216).

> G. References: appropriate credit to previous work?
> Some citations to references are suggested to add, but references are
> generally very appropriate. H. Clarity and context: lucidity of
> abstract/summary, appropriateness of abstract, introduction and
conclusions
>
> I think abstract, introduction, and conclusions are written well. In
> particular, the abstract presents fundamental properties of Dinkinesh,
such
> as diameter and the presence of satellite, and equatorial ridge. This is
> very good.